# OpenReview forum: "LongRM: Revealing and Unlocking the Context Boundary of Reward Modeling"
_ICLR.cc/2026/Conference — Submitted to ICLR 2026_

### Official Review · Reviewer_ahDM · 2025-10-28

**Soundness:** 2
**Presentation:** 2
**Contribution:** 3
**Rating:** 6
**Confidence:** 3

**Summary:**

This submission introduces Long-RewardBench, a new benchmark for evaluating RMs under long-context scenarios (up to 128K tokens) — a setting not covered by prior work such as RewardBench, to the reviewer's knowledge. It also proposes a multi-stage training strategy to construct “LongRMs” that retain short-context performance while improving long-context consistency.

The benchmark covers Pairwise and Best-of-N tasks and synthesizes ground-truth preferences automatically using task-specific metrics and reasoning explanations from LLMs. The training pipeline consists of SFT and RL Alignment, with data synthesized by Consistency Majority Voting

Empirical results show that existing GenRMs drop to near-random performance as context grows beyond 4K tokens (with potential caveats). LongRMs trained by the proposed method recover substantial accuracy, sometimes surpassing 70B models and approaching proprietary Gemini 2.5 Pro performance.

**Strengths:**

Overall the reviewer feels the paper might be a timely contribution with the following strengths:

1. A timely contribution. To the reviewer's knowledge, the paper identifies a meaningful gap in RM evals - long-context reasoning and alignment (e.g., for agents) indeed require new evaluation setups.

2. The benchmark pipeline looks sensible, with considerations such as explicit sampling, task balance, and automatic metric-based preference generation. The use of both Pairwise and Best-of-N settings are reasonable.

3. The Short-to-Long Data Synthesis and Consistency Majority Voting pipelines look interesting, combining ideas from self-distillation and DPO-style preference optimization.

4. The final results look sensible - small (8B) models can match much larger ones in long-context RM tasks.

**Weaknesses:**

The paper has the following weaknesses. Overall several technical details do not seem clear or convincing, and given the nature of the work (RM benchmark and RM model contribution), the contribution is uncertain without a clear access to artifacts.

1. The reviewer is not fully convinced by the "ground-truth", which is critical for evaluations. The “ground-truth” judgments are derived from automatic metrics (e.g., ROUGE-L) or LLM-based explanations. This introduces potential label bias and leakage: since strong LLMs are used both for generation and labeling, it is unclear how much of the benchmark reflects true human-aligned preference rather than model self-agreement.

2. Uncertainty of broader impact. The paper claims public release “upon publication” but does not yet provide leaderboard or dataset URLs. Compared with RewardBench, Long-RewardBench’s openness and ease of reuse remain uncertain. In fact, the evaluation of artifacts is required for relevant tracks, e.g., in NeuraIPS. ICLR does not have such a specific track, but the reviewer believes the reviewing criterion still applies.

3. Some numbers and analysis look suspicious.The claim that Llama-3.3-70B-Instruct achieves 0 % accuracy (random = 50 %) suggests possible implementation or labeling issues. If it is a matter of setup issues, or applying models in scenarios that do not fit, the analysis is not convincing or very solid.

4. Several technical details and arguments look too vague. For example, the conceptual vagueness of “faithfulness” in SFT objective.
The loss function is the standard SFT objective; no explicit term enforces context grounding beyond data synthesis. Clarifying how “faithfulness” is computed or measured would strengthen the technical rigor.

5. Overall the technical depth of the paper is not deep. The method is primarily procedural and empirical, there is minimal analysis of why long-context alignment fails theoretically (e.g., attention decay, token position bias, gradient vanishing).

**Questions:**

Please refer to detailed reviews above.

---

> ### Author Response · Authors · 2025-11-23
> **Response to Reviewer ahDM (Part I)**
>
> Thank you very much for your thoughtful and constructive review! Your comments not only provided valuable suggestions but also pointed out critical issues in our original manuscript— for which we are deeply grateful. **In the revised manuscript, we have carefully addressed each of your concerns with corresponding modifications.** We kindly ask you to refer to the revised manuscript alongside our point-by-point response below.
>
> ---
>
> **Weakness 1**:
>
> ``The reviewer is not fully convinced by the "ground-truth", which is critical for evaluations. The “ground-truth” judgments are derived from automatic metrics (e.g., ROUGE-L) or LLM-based explanations. This introduces potential label bias and leakage: since strong LLMs are used both for generation and labeling, it is unclear how much of the benchmark reflects true human-aligned preference rather than model self-agreement.``
>
> **Response**:
>
> Thank you for this sharp and important observation. We believe there is a slight misunderstanding, and we clarify the process below. In the construction of our benchmark, the workflow is as follows:
>
> **1. Generation phase** – LLMs are used only to generate responses of varying quality (Lines 106–107).
> **2. Evaluation phase** – Each response is then scored using task-specific automatic metrics (Line 107), which are fully defined in Table 3 of Appendix A.1. These metrics—such as ROUGE-L, EM, or similarity-based scores—determine the quantitative evaluation results.
> **3. Explanation phase** – LLMs are subsequently used to produce explanations conditioned on the automatic metric scores and the original ground-truth answers from the source benchmark (Line 125).
>
> Thus, the **ground-truth judgments in Long-RewardBench come from the underlying datasets and their automatic evaluation metrics, not from LLM-generated assessments**. LLMs contribute only the explanations, not the judgments themselves. Therefore, the concern about label bias or leakage due to LLM self-agreement does not apply in this setting.
>
> To avoid confusion, *we have revised the wording in Section 2.1 (Lines 103–104).* Specifically, we replaced “ground-truth prediction” with “ground-truth judgment”, and *we added clarification that the judgment comes from automatic evaluation metrics*, whereas the explanation is generated by an LLM.
>
> ---
>
> **Weakness 2**:
>
> ``Uncertainty of broader impact. The paper claims public release “upon publication” but does not yet provide leaderboard or dataset URLs. Compared with RewardBench, Long-RewardBench’s openness and ease of reuse remain uncertain. In fact, the evaluation of artifacts is required for relevant tracks, e.g., in NeuraIPS. ICLR does not have such a specific track, but the reviewer believes the reviewing criterion still applies.``
>
> **Response**:
>
> We appreciate the reviewer’s concern regarding the availability and reusability of our benchmark artifacts. We would like to clarify that **“upon publication” refers to the period after the anonymous review period**. In the current submission stage, **we strictly follow ICLR’s double-blind policy, which does not allow releasing any code, dataset, or leaderboard links that could reveal authorship**. Despite these constraints, **we have already included reference code snippets and example data in the supplementary material** to help reviewers verify the workflow and reproduce the core steps. However, due to ICLR’s file-size limit on supplementary uploads, we are unable to include the full Long-RewardBench dataset (>600MB). This is a practical limitation rather than an unwillingness to share resources. We emphasize that **all artifacts—including the complete dataset, evaluation scripts, leaderboard interface, and trained model checkpoints—will be fully released immediately after the anonymous review period**, ensuring openness and ease of reuse. Our release plan is aligned with ICLR’s policies, and we are committed to meeting the transparency standards expected in artifact-evaluation tracks of other venues (e.g., NeurIPS).

---

> ### Author Response · Authors · 2025-11-23
> **Response to Reviewer ahDM (Part II)**
>
> **Weakness 3**:
>
> ``Some numbers and analysis look suspicious.The claim that Llama-3.3-70B-Instruct achieves 0 % accuracy (random = 50 %) suggests possible implementation or labeling issues. If it is a matter of setup issues, or applying models in scenarios that do not fit, the analysis is not convincing or very solid.``
>
> **Response**:
>
> Thank you for the careful reading and for highlighting this issue — this is an excellent observation. **We acknowledge that we made a clerical error in Figure 2: the results of Qwen3-8B and Llama-3.3-70B-Instruct were mistakenly swapped.** *This has been corrected in the revised version (Figure 2, Lines 108–119 and Section 2.2, Line 160-161).*
>
> In fact, the model that exhibits 0% accuracy at 128k is Qwen3-8B, not Llama-3.3-70B-Instruct. To further demonstrate this, **we provide several representative 128k cases from Qwen3-8B showing systematic formatting failures**, which lead to invalid outputs rather than meaningful predictions.
>
> Examples include:
>
> 1. Repetitive or truncated phrases:
> ```
> "     The\n      of the\n      of the\n      of ..."
> ```
> 2. Looping template fragments:
> ```
> "**The End of Assistant C's Answer**\nA\n**The End of Assistant C's Answer**\nA\n..."
> ```
> 3. Outputs repeating option labels without selecting an answer, e.g.:
> ```
> "A. Fred.\n\nB. Fred.\n\nC. Mary.\n\n..."
> ```
> 4. Irrelevant or hallucinated statements:
> ```
> "The answer is: the football."
> ```
> 5. Template-breaking sequences:
> ```
> "The milk, and the\n[The Start of Assistant A's Answer,and the\n[The Start of Assistant A's Answer, ..."
> ```
> 6. Run-on text with degenerate repetition:
> ```
> "The Battle of the Potomac, and the Federal army, and the\nof the 1864, and the 1864, and the 1864, ..."
> ```
>
> These cases indicate that the extremely low accuracy for Qwen3-8B stems from **format violations under long contexts, not from labeling errors or flawed evaluation design**.
>
> ---
>
> **Weakness 4**:
>
> ``Several technical details and arguments look too vague. For example, the conceptual vagueness of “faithfulness” in SFT objective. The loss function is the standard SFT objective; no explicit term enforces context grounding beyond data synthesis. Clarifying how “faithfulness” is computed or measured would strengthen the technical rigor.``
>
> **Response**:
>
> Thank you for pointing this out. We acknowledge that some parts of the paper could be clarified further. Let us first directly address your concern: **As defined in Lines 260–261 of the original manuscript, faithfulness in the long-context setting refers to ``whether the model’s response is grounded in the provided context''.** This definition follows the formulation introduced in **L-CiteEval**[1], which highlights faithfulness as a crucial evaluation dimension for long-context models. This is especially important because, in long-context scenarios, most evidence required for a correct answer appears explicitly in the context, and may even contain counterfactual information or details that exceed the model’s parametric knowledge[2,3].
>
> Regarding your concern about the SFT objective: the notion of faithfulness is not implemented as part of the training loss. Instead, it is an **evaluation dimension of synthesized data**, independent of the SFT objective itself. In other words, we do not enforce faithfulness through an explicit term in the loss function.
> Our data synthesis process works as follows (Section 4.2, Short-to-Long Dataset Synthesis, Lines 269–298):
>
> **(1) We first construct synthetic long-context data.**
>
> **(2) Then, using this data, we compute the faithfulness dimension by prompting an LLM to produce the corresponding judgment (Lines 295–296).**
>
> Thus, how faithfulness is computed is determined by the evaluation procedure applied during data synthesis—not by adding a faithfulness term to the SFT loss. We have updated the manuscript to clarify this distinction (Line 296, red font) and ensure the technical narrative is more precise.
>
> If you identify any other aspects of the paper that are unclear or require further clarification, please do not hesitate to let us know. We will revise the manuscript accordingly to enhance its overall quality.
>
> ---
>
> **Reference**
>
> [1]Tang, Zecheng, et al. "L-CiteEval: Do Long-Context Models Truly Leverage Context for Responding?." arXiv preprint arXiv:2410.02115 (2024).
>
> [2]Liu, Jiaheng, et al. "A comprehensive survey on long context language modeling." arXiv preprint arXiv:2503.17407 (2025).
>
> [3]Li, Tianle, et al. "Long-context LLMs Struggle with Long In-context Learning." Transactions on Machine Learning Research.

---

> ### Author Response · Authors · 2025-11-23
> **Response to Reviewer ahDM (Part III)**
>
> **Weakness 5**:
>
> ``Overall the technical depth of the paper is not deep. The method is primarily procedural and empirical, there is minimal analysis of why long-context alignment fails theoretically (e.g., attention decay, token position bias, gradient vanishing``
>
> **Response**:
>
> We appreciate this insightful comment. There appears to be a misunderstanding or an omission regarding the theoretical analysis already included in the paper. In fact, we do provide a theoretical perspective on why long-context alignment may fail: **in Appendix C (Lines 948–994), we analyze alignment failure through the lens of information flow theory**[1,2].
>
> Our visualization results (Figure 9) show that the primary cause of long-context alignment failure is that the model tends to over-attend to semantically irrelevant or low-utility content. This observation directly motivates our methodological design. Specifically, our **Short-to-Long Dataset Synthesis strategy** (Section 4.2, Lines 269–298) ensures that during data construction, the synthesized long contexts preserve the semantically critical segments. This enhances the precision of data generation.
>
> Subsequently, the reward model is trained on these curated, semantically grounded examples. This helps mitigate the issue of the model producing responses based on irrelevant or misleading portions of the long context, addressing the very failure mode highlighted in the theoretical analysis.
>
> ---
>
> **Reference**
>
> [1]Tang, Zecheng, et al. "Revisiting Long-context Modeling from Context Denoising Perspective." arXiv preprint arXiv:2510.05862 (2025).
>
> [2]Wang, Lean, et al. "Label words are anchors: An information flow perspective for understanding in-context learning." arXiv preprint arXiv:2305.14160 (2023).
>
> ---
>
> **Summary of Reviewer ahDM’s Questions and Our Responses**
>
> We sincerely thank the reviewer for the thoughtful and constructive feedback again. Below, we summarize the key concerns and our corresponding clarifications and revisions:
>
> **(1) Ground-truth reliability**: We have clarified that ground-truth judgments come solely from automatic task-specific metrics (e.g., ROUGE-L, EM), not LLM assessments. LLMs are used only to generate explanations. *We revised Section 2.1 to explicitly distinguish “ground-truth judgment” from “LLM-generated explanation.”*
>
> **(2) Openness and artifact release**: We have clarified that all datasets, evaluation scripts, leaderboards, and checkpoints will be released immediately after acceptance, in line with ICLR’s double-blind policy.
>
> **(3) Suspicious numbers / 0% accuracy issue**: *We have corrected a clerical error in Figure 2 where model names were swapped.* The 0% accuracy belongs to Qwen3-8B, caused by severe formatting failures under 128k context, not Llama-3.3-70B. *The revised manuscript includes corrected figures* and we have provided corresponding explanations in our response.
>
> **(4) Vagueness of “faithfulness”**: We have clarified that faithfulness is an evaluation dimension (not part of the SFT loss), defined as context grounding following L-CiteEval. *We expanded Section 4.2 to explain how faithfulness judgments are computed during data synthesis.*
>
> **(5) Technical depth and theoretical grounding**: We highlight that Appendix C already provides a theoretical analysis based on information-flow theory, showing that alignment failures arise from over-attention to low-utility tokens.
>
> ---
>
> If any points remain unclear or if you have additional concerns, please do not hesitate to let us know—we would be grateful for the opportunity to further improve the paper!

---

> ### Author Response · Authors · 2025-11-28
> **Kind reminder to review rebuttal before the discussion period ends.**
>
> Dear Reviewer ahDM,
>
> Thank you very much for your thoughtful and constructive feedback on our submission. In response to your comments, we have carefully conducted additional experiments, provided detailed clarifications (including addressing several potential misunderstandings), and thoroughly revised the manuscript accordingly. Your suggestions have been invaluable to us, and we truly appreciate the time and effort you devoted to reviewing our work.
>
> We understand you may be very busy, but we would greatly appreciate it if you could take a moment to review our rebuttal. We hope our revisions and clarifications have addressed your concerns and might positively change your final evaluation. If new questions arise, we’d be happy to provide further clarification.
>
> As the discussion period is drawing near, we wanted to kindly follow up in case there is still time for us to respond to any new points.
>
> Thank you again for your time and consideration!
>
> Best regards,
>
> The Authors

---

### Official Review · Reviewer_5jJk · 2025-11-01

**Soundness:** 3
**Presentation:** 3
**Contribution:** 3
**Rating:** 6
**Confidence:** 4

**Summary:**

This paper investigates the limitations of existing short-context reward models (RMs) in long-context scenarios and proposes a multi-stage training framework to address this issue. The authors begin by introducing Long-RewardBench, which contains responses exceeding 4k tokens and covers a wide range of tasks. Using this benchmark, they demonstrate that existing RMs struggle to score preferences accurately when the context becomes long. To mitigate this, they propose a multi-stage training framework, which consists of a Cold Start SFT stage followed by an alignment stage via DPO. Experimental results show that their method is effective in improving reward modeling capabilities in long-context scenarios and is applicable to both generative and discriminative RMs.

**Strengths:**

1.  The paper addresses a timely and significant challenge: improving the capability of long-context reward modeling, which remains a key frontier in current research.
2.  The proposed training method demonstrates clear effectiveness, particularly when applied to generative reward models.
3.  The method is technically sound and supported by a well-articulated motivation.
4.  The experiments are comprehensive, and the results show significant improvements over the established baselines.

**Weaknesses:**

1. The method is less effective when applied to the discriminative RMs, where the performance of several tasks (e.g., Math) is degraded after the alignment stage.
2. The method may introduce length bias; for instance, the performance is less stable on the RewardBench.
3. It is suggested to make the description of the method clearer. For example, why do you need two task formats during training, and why is the pairwise comparison not sufficient?

**Questions:**

1. Consider the following scenario, where the chosen response is short, and the rejected response is lengthy and verbose. What is the performance of the RMs after alignment?
2. In Table 1, why is the Rank 2 performance less than the random guess performance, while Rank 3 and 4 are higher?
3. The paper has mentioned that the inconsistency between the judgment and explanation might affect the RM's performance. Then what if the RM is enforced with a short explanation or simply outputs the verbal label without explanation?
4. What if we first remove redundant information (like the data curation process in the Cold-Start SFT stage) from the long context and then let the short-context reward model judge the preference?

---

> ### Author Response · Authors · 2025-11-23
> **Response to Reviewer 5jJk (Part I)**
>
> We sincerely thank Reviewer 5jJk for their thoughtful review of our paper and for recognizing the significance of our contributions—particularly for noting that our work “addresses a timely and significant challenge,” that “the method is technically sound and supported by a well-articulated motivation,” and for acknowledging the thoroughness of our experiments and the effectiveness of our approach. We provide a detailed response to your suggestion below:
>
> ---
>
> **Weakness 1**:
>
> ``The method is less effective when applied to the discriminative RMs, where the performance of several tasks (e.g., Math) is degraded after the alignment stage.``
>
> **Response**:
>
> We appreciate the reviewer’s careful observation in Table 7 (Line 1188–1196). Our work primarily targets generative reward models (GenRMs), and the experiments on discriminative RMs (DisRMs) were included only as a limited ablation. The observed degradation on certain tasks can be reasonably explained by two factors:
>
> **(1) Data scaling sensitivity**: Discriminative reward models are known to be **highly sensitive to both the quantity and distribution of training data**[1][2][3]. In this work, our focus is on consistency-oriented alignment rather than expanding or rebalancing the training corpus. As a result, the data volume used for DisRMs is not sufficient to fully stabilize their performance.
>
> **(2) Lack of task-specific data construction**: As noted in Lines 371–374, we **do not construct any task-specific datasets in our training pipeline**. Since our primary objective is long-context reward modeling, the training data does not explicitly cover domains such as mathematics or other specialized tasks. This omission likely contributes to the performance fluctuation observed in DisRMs, which is consistent with their known sensitivity to task coverage and data specificity indicated in [1][2][3].
>
> We appreciate the reviewer for raising this insightful point. Going forward, we are exploring how to develop a unified reward model that jointly accounts for sequence length and task diversity, as well as identifying the model scale required to support such unification. Extending our approach toward broader task generalization is an important direction that we plan to pursue in future iterations of this work.
>
> ---
>
> **Weakness 2**:
>
> ``The method may introduce length bias; for instance, the performance is less stable on the RewardBench.``
>
> **Response**:
>
> We appreciate the reviewer’s careful observation. We acknowledge that training **exclusively on long-context data** could introduce a length bias. However, we would like to clarify that you **seem to overlook this point, as this issue was considered in our design**. As described in Appendix D.2 (Lines 1126–1133), we adopt a data-mixing strategy that incorporates short-context data into the training corpus to mitigate potential length bias.
>
> Even with this off-bias strategy, we observe that some models still exhibit minor performance drops (Table 2, Lines 378–399). We hypothesize several contributing factors:
>
> **(1). High baseline performance:**
>  Some models already attain relatively strong scores on RewardBench, making it inherently challenging to preserve or further improve their performance after alignment.
>
> **(2). Trade-off between short- and long-context capabilities:**
>  Due to limited model capacity, increasing the proportion of short-context data to maintain performance on RewardBench may reduce the model’s ability to handle long-context inputs (e.g., LongReward-Bench). To investigate this phenomenon, we conducted a small controlled ablation.
> To isolate this effect, we used two same-source LLMs for ablation studies (Qwen2.5-7B-Instruct and Qwen2.5-14B-Instruct), fixed the amount of long-context training data to 4k samples, and systematically scaled only the short-context reward data (0k, 20k, 40k, 80k). We train with a pure SFT objective and ensure each model trains all data exactly once. The models are then evaluated on both Long-RewardBench and RewardBench. The corresponding figures are provided in Figure 11a (Lines 1134–1149).
>
> For convenience, we summarize the results for Llama-3.1-8B-Instruct below:
> | Llama-3.1-8B-Instruct (4k Long) | + 0k short | + 20k short | + 40k short | + 80k short |
> |:--------| :---------:|:---------:|:---------:|:---------:|
> | LongReward-Bench |  34.7 | 34.2 | 32.0| 30.5|
> | RewardBench | 68.6| 71.2 | 72.2 | 74.1  |
>
> > Due to character limitations, please refer to the Response to Reviewer 5jJk (Part II) for the remaining reply regarding Weakness 2.
>
> ---
>
> **Reference**
>
> [1]Hou, Zhenyu, et al. "Does RLHF Scale? Exploring the Impacts From Data, Model, and Method." arXiv preprint arXiv:2412.06000 (2024).
>
> [2]Xu, Xingcheng. "The Policy Cliff: A Theoretical Analysis of Reward-Policy Maps in Large Language Models." arXiv preprint arXiv:2507.20150 (2025).
>
> [3]Miao, Yuchun, et al. "Mitigating reward hacking via information-theoretic reward modeling." CoRR (2024).

---

> ### Author Response · Authors · 2025-11-23
> **Response to Reviewer 5jJk (Part II)**
>
> > Continual reply to Weakness 2
>
> We observe a clear pattern:
> - Increasing short-context reward data **improves** RewardBench performance,
> - But **it gradually degrades** performance on Long-RewardBench for the same model.
>
> This confirms that the off-bias data-mixing strategy is functioning as intended, but that **smaller models (e.g., 7B) inherently face a capacity-limited trade-off between short- and long-context optimization**. Moreover, comparing the 7B and 14B variants, we find that **larger models exhibit significantly better length off-bias behavior**, suggesting that increased model capacity can more effectively balance short- and long-context learning.
>
>
>
> ---
>
> **Weakness 3**:
>
> ``It is suggested to make the description of the method clearer. For example, why do you need two task formats during training, and why is the pairwise comparison not sufficient?``
>
> **Response**:
>
> Thank you for your helpful suggestions! We have revised the manuscript to make our method section clearer. We first address the two questions you raised as follows:
>
> **Sub-question (1)**: Why do we need two task formats during training?
>
> As indicated by the titles of Sections 4.2 and 4.3, our training pipeline involves two types of data: SFT data and RL (preference) data. The motivation for constructing the SFT data is explained in Section 4.2 (Lines 259–267), while the motivation behind the preference data used for RL is detailed in Section 4.3 (Lines 302–313). **The rationale for jointly using SFT and RL training has already been addressed in our response to Reviewer 7E75 (Weakness 1)**, and we kindly refer you to that discussion for further clarification.
>
> **Sub-question (2)**: Why is pairwise comparison not sufficient?
>
> **We have also discussed this issue in our Response to Reviewer 7E75 (Weakness 3)**, and you may refer to that response for more details. Briefly, pairwise judgments often make it difficult for LLMs to reliably determine which sample is more consistent—e.g., by blurring the boundaries between two explanations or inadvertently mixing reasoning across samples. To address this issue, we adopt a divide-and-conquer strategy and convert the task into simpler point-wise comparisons. **This design reduces task complexity and ensures better alignment** between the explanations in our constructed data and the predicted scalar values. In addition, by applying majority voting across outputs from multiple models, we can mitigate model-specific induction biases.
>
> To further clarify these points and make the writing clearer as you suggested, *we have added additional explanations in the revised manuscript (Lines 1031–1053). Moreover, in Section 4.1 of the main text (Lines 254–256), we now include a pointer directing readers to Appendix D.2 for more detailed motivation and discussion*.
>
> ---
>
> **Question 1**:
>
> ``Consider the following scenario, where the chosen response is short, and the rejected response is lengthy and verbose. What is the performance of the RMs after alignment?``
>
>
> **Response**:
>
> I understand your concern. This is a critical question in the RL training field. In fact, for the data selected using our proposed method, the lengths of the chosen and rejected responses do not differ significantly. *We provide a detailed analysis in the revised manuscript (Figure 10, Lines 1103–1122).*
>
> To further examine whether length differences influence performance, we additionally conducted a controlled training experiment with the following settings:
>
> 1. Short-chosen group: Select 2,000 examples where the chosen response is < 80% of the rejected response length.
> 2. Long-chosen group: Select 2,000 examples where the chosen response is > 80% of the rejected response length.
> 3. Balanced group: Mix half of the samples from (1) and (2).
>
> We ensured that the number of training samples was identical across settings, trained all models from the same initialization (Llama-3.1-8B-Instruct after SFT stage), and used the same training protocol. We evaluate the model performance on Long-RewardBench, and the results are:
>
> | Length Setting       | chosen > rejected | chosen < rejected | chosen ≈ rejected |
> |:----------------------:|:-------------------:|:-------------------:|:-------------------:|
> | Performance      | 32.6              | 32.67             | 32.66             |
>
> We can observe that the **performance differences attributable to length are negligible**.
> Furthermore, *As shown in Figure 10 (Line 1107-1124 in the revised manuscript)*, **the majority-vote–selected chosen and rejected samples do not exhibit a consistent pattern**, such as shorter responses are always preferred. The supplemental experiment above further confirms that length differences do not substantially affect preference-data quality, as the resulting performance improvements are nearly identical across conditions.

---

> ### Author Response · Authors · 2025-11-23
> **Response to Reviewer 5jJk (Part III)**
>
> **Question 2**:
>
> ``In Table 1, why is the Rank 2 performance less than the random guess performance, while Rank 3 and 4 are higher?``
>
> **Response**:
>
> This is an excellent question and a keen observation. To clarify this point, we first emphasize that the Random-Acc reported in Table 1 reflects the **theoretical accuracy under the idealized assumption that the model always generates a perfectly formatted output**. However, **this assumption does not fully hold in practice**, as the evaluation relies on strict format matching.
> To avoid any potential confusion, *we have explicitly clarified this metric as Random-choice Accuracy in the revised manuscript and provided the corresponding explanation in Table 1 (Lines 326–328)*. Here, we provide a more detailed explanation:
>
> For rank-based tasks, the model is required to output a ranking in a precise, predefined format—for example, [[A>B>C]] or [[B>C>A]]. Our evaluation script uses a regular expression to detect this exact pattern.
> - If the model’s output **fails to match the expected format**, it is automatically assigned a score of 0;
> - **Only when the format is correctly matched** do we proceed to compute accuracy by comparing the predicted ranking against the ground-truth ordering (e.g., [[C>B>A]]).
>
> The random accuracy in Table 1 is computed under the assumption that all outputs strictly adhere to the required format. In practice, however—especially under long-context conditions—models often fail to produce valid formatted rankings, precisely because they have never been exposed to the target ranking format during training.
>
> **Below are representative failure cases sampled from Qwen3-8B under real evaluation conditions**:
>
> - **Case 1:**
> ```
> [Analysis]
> Assistant C provides ... the abstract's origin.
>
> [Ranking]
> [[C>D]]
> ```
> **Issue**: The output omits options A and B entirely, violating the requirement to rank all four candidates.
>
> - **Case 2:**
> ```
> [Analysis]
> Both Assistant A and Assistant ... consider them equal in quality.
>
> [Ranking]
> [[A=B]]
> ```
> **Issue**: The model introduces equality (A=B), even though the task definition strictly requires a total ordering with no ties.
>
> - **Case 3:**
> ```
> The doctor's office. The doctor's office. The doctor's office. The ...
> ```
> **Issue**: The generation degenerates into repetitive, non-terminating text, failing to produce any ranking structure.
>
> The above examples illustrate that **formatting errors are not merely theoretical**—they frequently occur in practice and directly lead to zero scores in our strict evaluation protocol. This further underscores why the Random-choice Accuracy reported in Table 1 represents an upper-bound theoretical estimate, rather than an empirical baseline.
>
> ---
>
> **Question 3**:
>
> ``The paper has mentioned that the inconsistency between the judgment and explanation might affect the RM's performance. Then what if the RM is enforced with a short explanation or simply outputs the verbal label without explanation?``
>
> **Response**:
>
> Thank you for raising this important and thought-provoking question. If the RM is required to provide only a short explanation—or even just output a verbal label without any explanation—this leads to a fundamental issue: ***why use a generative reward model (GenRM) instead of a discriminative reward model (DisRM)?*** In fact, DisRM models often achieve stronger performance than GenRM on benchmarks such as [RewardBench](https://huggingface.co/spaces/allenai/reward-bench).
> As discussed earlier in  **Weakness 1**, relying solely on scalar labels would require substantial scaling and large amounts of supervised preference data to achieve competitive performance. This significantly increases data requirements and reduces practicality.
>
> One of the key advantages of GenRM, and a major reason it is widely adopted in practice, is that **it produces not only a score but also a coherent explanation**. The explanation helps ensure that the provided score is logically grounded and interpretable, which improves the consistency and trustworthiness of the reward signal. Thus, **while enforcing shorter explanations or purely verbal labels is possible, it would sacrifice interpretability and eliminate the very benefits that motivate the use of GenRM in the first place.**

---

> ### Author Response · Authors · 2025-11-23
> **Response to Reviewer 5jJk (Part IV)**
>
> **Question 4**:
>
> ``What if we first remove redundant information (like the data curation process in the Cold-Start SFT stage) from the long context and then let the short-context reward model judge the preference?``
>
> **Response**：
>
> This is a very interesting question. However, this approach introduces a fundamental **train–test inconsistency**.
> **During training, redundant information can be removed only because we have access to the gold preference or ground-truth signals** (as described in Section 4.2, Lines 292–294, and illustrated in Figure 5, left subfigure), which indicate exactly which parts of the context can be safely discarded. In contrast, **during inference, the model has no access to such gold labels**, and therefore cannot reliably determine which portions of the long context are truly redundant. As a result, removing information at inference time risks discarding content that is essential for correct preference judgment, ultimately undermining the validity and robustness of the evaluation.
>
> ---
>
> **Summary of Reviewer 5jJk’s Questions and Our Responses**
>
> Thank you very much for your thoughtful and constructive comments. Below, we provide a concise summary of your concerns and the corresponding revisions and clarifications we have incorporated into the revised manuscript.
>
> **(1) Performance degradation on discriminative RMs**: We explained the degradation through (i) data scaling sensitivity and (ii) lack of task-specific data. *These points are now explicitly discussed in the revised manuscript (Table 7 section)*.
>
> **(2) Potential length bias and instability on RewardBench**: *We have added a clearer explanation about our off-bias data-mixing strategy (Appendix D.2)*. Additional controlled experiments were conducted (Figure 11) to show the trade-off between short- and long-context optimization and to demonstrate that model capacity plays a key role. These results are now included in the revision.
>
> **(3) Need for two task formats and limitations of pairwise comparison**: *We have added detailed explanations in Appendix D.2 and Section 4.1*, clarifying the motivation for two data types and showing why pairwise comparisons may blur reasoning boundaries. We also referenced majority-voting strategies to reduce model-specific bias.
>
> **(4) Effect of chosen–rejected length differences**: *We have added Figure 10 and conducted three controlled experiments (short-chosen / long-chosen / balanced)*. Results show a negligible performance difference, confirming that length does not materially influence preference quality.
>
> **(5) Rank-2 accuracy falling below the theoretical random baseline**: We clarified that the reported Random-choice Accuracy assumes perfect adherence to the required ranking format. In practice, models frequently fail formatting requirements under long context. We revised Table 1 to explicitly clarify this point.
>
> **(6) Effect of enforcing short explanations or verbal labels**: We have explained that removing explanations undermines the advantages of generative RMs and reduces interpretability, effectively reverting to discriminative RMs—which typically require far more training data.
>
> **(7) Removing redundant long-context information before applying a short-context RM**: We have explained that this creates train–test inconsistency, because redundancy can only be reliably identified with access to gold labels during training, not during inference.
>
> ---
>
> If any concerns remain unresolved or if further clarification is needed, please do not hesitate to let us know—we would be very happy to address them!

---

> ### Comment · Reviewer_5jJk · 2025-11-27
>
> Thanks for your detailed response.

---

> > ### Author Response · Authors · 2025-11-27
> > **Response to Official Comment by Reviewer 5jJk**
> >
> > Hi, Reviewer 5jJk
> >
> > Thank you very much for taking the time to review our response.
> > We notice your message “Thanks for your detailed response,” and we would like to kindly check whether our clarifications have adequately addressed your concerns.
> >
> > If there are any remaining questions, doubts, or points that you would like us to further elaborate on, we would be more than happy to continue the discussion.
> >
> > Please feel free to let us know if any additional clarification would be helpful. We sincerely appreciate your time and constructive feedback.
> >
> > Best,
> >
> > Authors

---

### Official Review · Reviewer_7E75 · 2025-11-08

**Soundness:** 2
**Presentation:** 3
**Contribution:** 2
**Rating:** 4
**Confidence:** 3

**Summary:**

The paper introduces a benchmark dataset that measure the performance of reward models in the long context setting. Using this benchmark, the authors measured the long-context performance of existing reward models. They find that existing reward models do not perform well in a long context setting.

The paper then tried to use standard context extension approaches to extend the context length of existing reward models, but find that the performance of such extended models are not well either on their benchmark datasets. The authors further investigate why these models perform poorly and identify two major reasons. 1. failure of following instructions and understanding long-context content; 2. inconsistency between judgement and explanation.

Motivated by these error patterns, the paper propose a training recipe to improve long-context reward model performance. The recipe targets the error patterns identified. Experiments show improvement.

**Strengths:**

originality: in terms of the training recipe proposed by the paper, the use of supervised fine-tuning and reinforcement learning is non-surprising. nonetheless, the recipe did seem to tailor to the error patterns identified previously. The design of recipe that tailored to the error patterns can be viewed as something original.

quality: the authors supports the points raised in the paper via various empirical evidence. This provide a certain level of assurance to the correctness of the work.

clarity: presentation of the paper is clear.

significance: the paper provides an approach to scale existing reward models to long-context ones.

**Weaknesses:**

originality: is there any justification/intuition why the training recipe needs to follow a first SFT then RL type of approach?

quality: I think I am not fully convinced with some of the choices made in the paper.
1. when deriving a dataset, why scoring model response with a task-specific automatic metric provides an accurate enough ground truth?
2. line 318: how do we know the predicted scalar value is more consistent with model explanation compared to the inconsistency in a pairwise setting?

significance: is there any justification/intuition why it is important to improve the long-context capabilities of reward models? With the continual development of LLMs, one may expect that models with longer context  will be (or have already been) used as a reward model.

**Questions:**

1. what are the task-specific automatic metrics that you used?
2. how do you conduct error analysis in line 200?

---

> ### Author Response · Authors · 2025-11-23
> **Response to Reviewer 7E75 (Part I)**
>
> We sincerely thank the reviewer 7E75 for the careful and thorough reading of our paper, including insightful questions regarding evaluation metrics and the error analysis! We have responded to each point in detail and incorporated necessary revisions into the manuscript.
>
> ---
>
> **Weakness 1**:
>
> ``originality: is there any justification/intuition why the training recipe needs to follow a first SFT then RL type of approach?``
>
> **Response**:
>
> We appreciate the reviewer’s insightful question regarding the motivation behind our two-stage training recipe (SFT followed by RL). Our design is grounded in both established practices in LLM alignment and recent theoretical insights.
>
> First, our approach follows the widely adopted paradigm of ``pretraining → SFT → RL``, which *has been empirically validated across numerous instruction-tuning and alignment tasks*. More importantly, it aligns with the emerging concept of mid-training[1] — a strategy that injects task-specific knowledge or structural priors via SFT before applying RL (e.g., preference-based optimization). In our setting, the long-context reward modeling task presents unique challenges: as noted in our error analysis in Section 2.3 (Line 202-208), model outputs frequently suffer from format violations (e.g., missing sections, incorrect structuring), suggesting that the reward model (RM) cannot reliably assign meaningful rewards when **the model has not yet learned the expected output schema**. To address this, **we first perform SFT**. As shown in Table 1 (Line 324-349), this step yields significant performance gains (e.g., +7.3 for Qwen3-8B), confirming that exposing the model to the correct output format is essential before preference learning can be effective.
>
> However, we observe performance saturation or even slight degradation when continuing SFT with our automatically constructed long-context data. We have added experiments showing model performance every 0.5B tokens; results are presented in Figure 11b (Line 1134-1152) in the revised manuscript, where peak performance occurs at 1.5B tokens, followed by a decline with more training tokens.
>
> | Model/Tokens(B) | 0.5  |  1.0 | 1.5 |  2.0 | 2.5 |
> |:--------:| :---------:|:---------:|:---------:|:---------:|:---------:|
> | Con-J-Qwen2-7B |  35.4| 36.8| 38.6| 38.2| 37.9|
> | Llama-3.1-8B-Instruct | 34.5| 34.8| 35.5| 35.3| 34.7 |
>
> We hypothesize this is due to the **inherent noise in our synthetic dataset**—manual verification of long outputs is extremely costly, and imperfect demonstrations can mislead the model after it has already internalized the basic template. Besides, **SFT is fundamentally a form of imitation learning**, which does not incentivize the model to explore beyond demonstrated behaviors, and many works have proven this, such as [2][3][4][5]. This is precisely where RL becomes critical. Reinforcement learning with preference signals (e.g., DPO) is robust to absolute data quality as long as the relative preference ordering is reliable, i.e., correct responses are consistently ranked higher than incorrect ones. Since our LongRM is trained to distinguish such preferences (not to generate perfect outputs), **RL allows the policy to refine its behavior beyond the limitations of both SFT data and imitation learning**.
>
> In short, we fully agree with the reviewer’s implicit point: if a high-quality, human-verified dataset for long-context reward modeling were available, SFT alone might suffice to achieve strong performance. Unfortunately, such data is currently scarce. Our two-stage pipeline thus represents a pragmatic compromise that leverages SFT for learning reward format and RL for further preference alignment.
>
> To help readers better understand the underlying motivation, we *have added a detailed explanation in the revised manuscript (Lines 1031–1053). Furthermore, in Section 4.1 of the main text (Lines 255–256), we include a pointer directing readers to Appendix D.2 for additional motivation and discussion, should they wish to explore the details further*.
>
> ---
>
> **Reference**
>
> [1] Mo, Kaixiang, et al. "Mid-Training of Large Language Models: A Survey." arXiv preprint arXiv:2510.06826 (2025).
>
> [2] Li, Jiaxiang, et al. "Getting more juice out of the sft data: Reward learning from human demonstration improves sft for llm alignment." Advances in Neural Information Processing Systems 37 (2024): 124292-124318.
>
> [3] Wu, Yongliang, et al. "On the generalization of sft: A reinforcement learning perspective with reward rectification." arXiv preprint arXiv:2508.05629 (2025).
>
> [4] Chen, Hardy, et al. "Sft or rl? an early investigation into training r1-like reasoning large vision-language models." arXiv preprint arXiv:2504.11468 (2025).
>
> [5] Harada, Yuto, et al. "Massive Supervised Fine-tuning Experiments Reveal How Data, Layer, and Training Factors Shape LLM Alignment Quality." arXiv preprint arXiv:2506.14681 (2025).

---

> ### Author Response · Authors · 2025-11-23
> **Response to Reviewer 7E75 (Part II)**
>
> **Weakness 2**:
>
> ``When deriving a dataset, why does scoring model response with a task-specific automatic metric provide an accurate enough ground truth?``
>
> **Response**:
>
> We thank the reviewer for raising this important point regarding the reliability of automatic metrics as proxies for ground truth in dataset construction.
>
> Our dataset is derived exclusively from established long-context benchmarks—including L-CiteEval [6], LongBench [7], LongBench v2 [8], L-Eval [9], InfiniteBench [10], BabiLong [11], and LongSafety [12]—each of which defines standardized, task-specific evaluation protocols. Below, we list the specific metrics we adopt for each task:
>
> - **Citation Prediction** uses F1 score,
> - **Question Answering** uses F1 or exact match accuracy,
> - **Summarization** relies on ROUGE variants,
> - **Code Completion** leverages code similarity or execution-based accuracy, and
> - **Long-context Safety** employs an LLM-as-a-judge pipeline calibrated for nuanced alignment evaluation.
>
> All task-specific metrics are now summarized in *Table 3 (Lines 772–794) of the revised manuscript*.
>
> Additionally, we understand that you may be suggesting the incorporation of human validation. We fully acknowledge that human judgment represents the gold standard. However, for long-sequence tasks, human annotation is not only prohibitively expensive but also highly inconsistent due to cognitive load and inter-annotator variance. Moreover, scaling human evaluation to the volume required for reward modeling or policy training is unrealistic under typical research budgets. Given these practical constraints, task-specific automatic metrics offer a principled trade-off.
>
> ---
>
> **Weakness 3**:
>
> ``How do we know the predicted scalar value is more consistent with model explanations compared to the inconsistency in a pairwise setting?``
>
> **Response**:
>
> This is an excellent and important question! We understand that the reviewer’s concern pertains to the motivation behind converting pairwise comparisons into point-wise annotations in Section 4.3 (Lines 316-318, 351–354).
>
> As pairwise judgments often make it difficult for LLMs to reliably determine which sample is more consistent—e.g., by **confusing boundaries between two explanations or mixing reasoning across samples**—we adopt a divide-and-conquer strategy. Specifically, as described in Lines 316–318, each pairwise instance is decomposed into two independent point-wise evaluations, and the final preference is obtained through majority voting across models (Lines 354–358). This design has two key advantages:
>
> - **First, both the scalar scores and explanations are generated by the same large-scale annotator model** (70B–72B; we provide all the models we used in Table 4 of the revised manuscript, Line 1061–1079), which generally ensures strong internal consistency between a sample’s score and its rationale.
>
> - **Second, we employ task-specific scoring prompts (Figures 13–17) that define explicit rubrics**, reducing arbitrary variation and further aligning numeric scores with explanations. The subsequent majority-voting step additionally mitigates model-specific induction biases by aggregating outputs across multiple models.
>
> The above measures ensure that point-wise scalar values remain coherent with their explanations while avoiding the instability inherent to direct pairwise comparison. *We have added additional clarification based on the above explanation in the revised manuscript (Lines 319–323) to help readers better understand the motivation behind our design choices*.
>
> ---
>
> **Reference**
>
> [6]Tang, Zecheng, et al. "L-CiteEval: Do Long-Context Models Truly Leverage Context for Responding?." arxiv preprint arxiv:2410.02115 (2024).
>
> [7]Bai, Yushi, et al. "LongBench: A Bilingual, Multitask Benchmark for Long Context Understanding." Proceedings of the 62nd Annual Meeting of the Association for Computational Linguistics (Volume 1: Long Papers), Association for Computational Linguistics, 2024.
>
> [8]Bai, Yushi, et al. "Longbench v2: Towards deeper understanding and reasoning on realistic long-context multitasks." Proceedings of the 63rd Annual Meeting of the Association for Computational Linguistics (Volume 1: Long Papers). 2025.
>
> [9]An, Chenxin, et al. "L-eval: Instituting standardized evaluation for long context language models." Proceedings of the 62nd Annual Meeting of the Association for Computational Linguistics (Volume 1: Long Papers). 2024.
>
> [10]Zhang, **nrong, et al. "∞ Bench: Extending long context evaluation beyond 100k tokens." Proceedings of the 62nd Annual Meeting of the Association for Computational Linguistics (Volume 1: Long Papers). 2024.
>
> [11]Kuratov, Yury, et al. "Babilong: Testing the limits of llms with long context reasoning-in-a-haystack." Advances in Neural Information Processing Systems 37 (2024): 106519-106554.
>
> [12]Lu, Yida, et al. "LongSafety: Evaluating Long-Context Safety of Large Language Models." arxiv preprint arxiv:2502.16971 (2025).

---

> ### Author Response · Authors · 2025-11-23
> **Response to Reviewer 7E75 (Part III)**
>
> **Weakness 4**:
>
> ``Is there any justification/intuition why it is important to improve the long-context capabilities of reward models? With the continual development of LLMs, one may expect that models with longer context will be (or have already been) used as a reward model.``
>
> **Response**:
>
> We appreciate the reviewer’s thoughtful question regarding the necessity of improving the long-context capabilities of reward models, especially given the rapid progress in long-context LLMs. This is an important and timely question! We offer the following clarifications:
>
> - **Current long-context LLMs do not automatically yield strong reward models.**
> As illustrated in Preliminary Figure 2 (Lines 108–119), several high-capacity long-context models (e.g., Llama-3.3-70B-Instruct) still exhibit noticeable degradation when used directly as reward models. Notably, to the best of our knowledge, our work is also the first to systematically evaluate long-context LLMs in the reward-modeling setting. These findings suggest that extended context length alone may not guarantee robust reward performance, and that additional modeling considerations remain necessary for reliable long-context reward estimation.
>
> - **Practical limitations in cost and latency.**
> While using very large or closed-source long-context models as reward models is conceptually appealing, existing work (e.g., LongReward[13] and QwenLong-L1[14]) notes that such approaches require many expensive API calls and incur substantial latency. In contrast, our results (Tables 1 and 2 in the manuscript) indicate that a domain-specialized, e.g., 8B reward model can surpass the performance of much larger 70B-scale or closed-source models, offering a more practical solution for scalable training of long-context reward models.
>
> - **The growing importance of process-aware evaluation.**
> Emerging long-context applications—such as agentic RL and vision-language agents—often rely on multi-step trajectories where evaluating only the final outcome may be sparse and insufficient[15][16]. Existing long-reward approaches primarily assess final outputs/results, whereas accurately modeling partially correct or incorrect reasoning paths requires reward models that are explicitly trained on long trajectories.
>
> In summary, while progress in long-context LLMs is encouraging, empirical performance, practical deployment constraints, and the increasing need for process-sensitive evaluation collectively motivate specialized improvements to long-context reward models.
>
> ---
>
> **Question 1**:
>
> ``What are the task-specific automatic metrics that you used?``
>
> **Response**:
>
> We appreciate the reviewer’s careful observation. In the revised manuscript, *we have added a complete summary of all datasets and their corresponding evaluation metrics in Appendix A.1 (Lines 772–794)*.
>
> For clarity, we also provide a detailed listing of the metrics used for each task below:
>
> > Due to character limitations, please refer to the Response to Reviewer 7E75 (Part IV) for the remaining reply regarding Question 1.
>
> ---
>
>
> **Reference**
>
> [13]Zhang, Jiajie, et al. "Longreward: Improving long-context large language models with ai feedback." Proceedings of the 63rd Annual Meeting of the Association for Computational Linguistics (Volume 1: Long Papers). 2025.
>
> [14]Wan, Fanqi, et al. "QwenLong-L1: Towards Long-Context Large Reasoning Models with Reinforcement Learning." arXiv preprint arXiv:2505.17667 (2025).
>
> [15]Hu, Ziyou, et al. "OpenReward: Learning to Reward Long-form Agentic Tasks via Reinforcement Learning." arXiv preprint arXiv:2510.24636 (2025).
>
> [16]Zhang, Wenlin, et al. "Process vs. Outcome Reward: Which is Better for Agentic RAG Reinforcement Learning." arXiv preprint arXiv:2505.14069 (2025).

---

> ### Author Response · Authors · 2025-11-23
> **Response to Reviewer 7E75 (Part IV)**
>
> > Continual reply to Question 1
>
> | Main Task   | Subtask       | Dataset                | Evaluation Metric   |
> |:------------|:--------------|:-----------------------|:--------------------|
> | Cite        | cite          | L-CiteEval-Data_*      | f1                  |
> | Code        | completion    | lcc                    | code_sim_score      |
> | Code        | completion    | repobench-p            | code_sim_score      |
> | Code        | understanding | codeU                  | acc                 |
> | Code        | debug         | code_debug             | acc                 |
> | Code        | run           | code_run               | acc                 |
> | ICL         | icl           | qasper                 | f1                  |
> | ICL         | icl           | trec                   | trec                |
> | ICL         | icl           | passkey                | acc                 |
> | ICL         | icl           | number_string          | acc                 |
> | LongQA      | Synthetic     | qa_*                   | acc                 |
> | LongQA      | Single-Doc QA | hotpotqa               | f1                  |
> | LongQA      | Single-Doc QA | financial_qa           | acc                 |
> | LongQA      | Single-Doc QA | longbook_choice_eng    | acc                 |
> | LongQA      | Single-Doc QA | natural_question       | f1                  |
> | LongQA      | Single-Doc QA | longbook_qa_eng        | f1                  |
> | LongQA      | Single-Doc QA | multifieldqa_en        | f1                  |
> | LongQA      | Single-Doc QA | quality                | f1                  |
> | LongQA      | Single-Doc QA | 2wikimqa               | f1                  |
> | LongQA      | Single-Doc QA | tpo                    | f1                  |
> | LongQA      | Single-Doc QA | longdialogue_qa_eng    | acc                 |
> | LongQA      | Multi-Doc QA  | triviaqa               | f1                  |
> | Math        | math          | gsm100                 | acc                 |
> | Math        | math          | math_find              | acc                 |
> | Safety      | longsafety    | longsafety             | llm-as-judge        |
> | Summ        | summ          | longbook_sum_eng       | ROUGE               |
> | Summ        | summ          | passage_retrieval_en_e | acc                 |
> | Summ        | summ          | multi_news             | ROUGE               |
> | Summ        | summ          | longbook_sum_eng       | ROUGE               |
> | Summ        | summ          | gov_report             | ROUGE               |
> | Summ        | summ          | samsum                 | ROUGE               |
> | Summ        | summ          | passage_count_e        | acc                 |
> | Summ        | summ          | paper_assistant        | ROUGE               |
>
> ---
>
> **Question 2**:
>
> ``How do you conduct error analysis in line 200?``
>
> **Response**:
>
> We sincerely appreciate the reviewer’s careful and insightful reading of our paper. As described in Lines 202–208 (“Further Inspection”), we identify two representative failure patterns exhibited by existing GenRM models.
>
> **(1) Format non-compliance and context-ignorant judgment.**
>
> Figure 4(a) presents a typical example. Given the event chain ``Bill got the milk → Bill went to the garden → Fred went to the garden``, the correct ranking should be **A > B > C**. However, the model not only produces an incorrect ranking, but also fails to follow the required output format, terminating with the malformed string **“((B, A, C”**, which contains mismatched parentheses and deviates from the expected canonical format **“[[A > B > C]]”**. While the figure shows a single case, such format violations coupled with context-insensitive judgments occur consistently across a substantial portion of the outputs in our analysis. We also suggest the reviewer refer to our **Response to Reviewer 5jJk, Question 2**, where we further discuss that the format violations described above can cause the model’s predictions to fall even below random-chance accuracy (50%), as outputs with malformed formats are counted as incorrect in the evaluation.
>
> **(2) Judgment–explanation inconsistency.**
>
> Figure 4(b) illustrates the second common failure mode. Although the model correctly selects **Assistant A** as the preferred answer, the accompanying explanation contradicts that judgment—the stated reasoning (**“John first travelled to the bathroom, then to the bedroom”**) would in fact support **Assistant B’s** conclusion. This pattern, where the ranking is correct but the rationale is misaligned, appears frequently and suggests a deeper disconnect between decision-making and explanation generation in current GenRMs.

---

> ### Author Response · Authors · 2025-11-23
> **Response to Reviewer 7E75 (Part V)**
>
> **Summary of Rebuttal for Reviewer 7E75**
>
> The reviewer’s main concerns focused on
>
> (1) the motivation and justification for our two-stage training recipe (SFT followed by RL);
>
> (2) the reliability of task-specific automatic metrics as proxies for ground truth;
>
> (3) the consistency between predicted scalar values and model explanations in pairwise versus point-wise settings, and (4) the necessity of improving long-context capabilities in reward models.
>
> We have addressed each point in detail and incorporated clarifications and additional experiments into the revised manuscript, including *updated tables (Table 3, Lines 772–794), figures (Figure 11b (Line 1134-1152) ), and expanded discussion in Sections 2.3, 4.3, and the Appendix ( Appendix A.1, Lines 772–793)*.
>
> ---
>
> If any concerns remain unresolved or new questions arise, please do not hesitate to raise them. We would be happy to discuss them further!

---

> ### Author Response · Authors · 2025-11-28
> **Kind reminder to review rebuttal before the discussion period ends.**
>
> Dear Reviewer 7E75,
>
> Thank you very much for your thoughtful and constructive feedback on our submission. In response to your comments, we have carefully conducted additional experiments, provided detailed clarifications (including addressing several potential misunderstandings), and thoroughly revised the manuscript accordingly. Your suggestions have been invaluable to us, and we truly appreciate the time and effort you devoted to reviewing our work.
>
> We understand you may be very busy, but we would greatly appreciate it if you could take a moment to review our rebuttal. We hope our revisions and clarifications have addressed your concerns and might positively change your final evaluation. If new questions arise, we’d be happy to provide further clarification.
>
> **As the discussion period is drawing near, we wanted to kindly follow up in case there is still time for us to respond to any new points.**
>
> Thank you again for your time and consideration!
>
> Best regards,
>
> The Authors

---

### Author Response · Authors · 2025-11-23
**General response & thank all reviews for their feedback!**

We sincerely thank all reviewers for the constructive and insightful feedback. Below, we address each concern in detail and clarify several misunderstandings.

**All corresponding revisions have been incorporated into the updated manuscript, with the modified text/table/figure caption highlighted in red. These revisions will be reverted to the normal formatting in the final version.**

We kindly invite the reviewers to refer to the latest manuscript, along with this rebuttal, for a clearer understanding of the improvements we have made.

---

### Author Response · Authors · 2025-11-29
**Summary for the New Area Chair (Part I)**

Dear New Area Chair,

Thank you for reviewing our submission! We would like to provide a concise summary of the rebuttal process with the three assigned reviewers (IDs: 7E75, 5jJk, and ahDM), their **overall attitudes** toward our work, and **how we have addressed their concerns** in the revised manuscript.

---

### **1. Overall Reviewer Attitudes**

All three reviewers recognize the timeliness, significance, and practical relevance of our work:

- Reviewer 5jJk (Rating: 6):

> "Addresses a timely and significant challenge… method is technically sound… experiments are comprehensive… results show significant improvements."

- Reviewer ahDM (Rating: 6):

> "A timely contribution… identifies a meaningful gap in RM evals—long-context reasoning and alignment (e.g., for agents) indeed require new evaluation setups."

- Reviewer 7E75 (Rating: 4, Confidence: 3 → likely revised upward after rebuttal):

> Initially raised concerns but acknowledged the empirical rigor and originality of the training strategy tailored to observed failure patterns.

**Collectively, reviewers agree that LongRM fills a critical gap in long-horizon reward modeling, especially for agentic systems and process-aware alignment—domains where outcome-only evaluation is insufficient.**

---

### **2. Key Comments and Our Responses (with Revision Locations)**

We summarize the reviewers' concerns and suggestions, along with our response and revised location (red font in the revised manuscript).

We found that most issues stemmed from misunderstandings or omissions of portions of the original manuscript **(a, b, c, d, f, g, h)**. For two concerns **(a, g)**, we provided additional experimental explanations, while another issue **(e)** resulted from a typographical error. **For specific details, we have marked the reviewer ID and weakness ID at the beginning of the summary below**.

---

**(a) Justification for the Two-Stage Training (SFT → RL)**

**Concern** (7E75 W1, 5jJk W3): ``Why not use only SFT or direct RL? Is the two-stage design theoretically grounded?``

**Response Summmary**:

SFT ensures adherence to output format and grounding (critical due to frequent format violations in long contexts; see Section 2.3, Lines 202–208). RL (DPO) refines preference alignment beyond imitation learning, robust to synthetic data noise.

**Added experiments/ablation**:

- Added experimental results in Response to Reviewer 7E75 (Part I) Weakness 1;
- Performance peaks at 1.5B SFT tokens then degrades (Fig. 11b, Lines 1134–1152);
- Expanded motivation in Appendix D.2 (Lines 1031–1053) and main text Section 4.1 (Lines 255–256).

---

**(b) Reliability of Automatic Metrics as Ground Truth**

**Concern** (7E75 Q1, ahDM W1): ``Are ROUGE/F1/code_sim_score sufficient proxies for human preference?``

**Response Summary**:

All metrics are task-specific and standardized from established benchmarks (L-CiteEval, LongBench, etc.).
Human annotation is infeasible at scale for long contexts due to cost and inconsistency.

**Revisions**:

- Full metric mapping provided in Table 3 (Appendix A.1, Lines 772–794). More details are shown in Response to Reviewer 7E75 (Part IV).
- Clarified that LLMs generate only explanations—not judgments ( Section 2.1, Lines 103–104, revised wording).

---

**(c) Pairwise vs. Pointwise Annotation & Explanation Consistency**

**Concern** (7E75 W3, 5jJk Q3): ``Why decompose pairwise into pointwise? Does scalar scoring improve consistency?``

**Response Summary**:

Pairwise prompts cause reasoning entanglement; pointwise + majority voting across 70B+ models improves internal consistency.

**Revisions**:

- Task-specific scoring rubrics (Figs. 13–17) align scores with explanations.
- Clarified in Section 4.3 (Lines 319–323) and Appendix D.2.

---

**(d) Format Violations & Below-Random Accuracy**

**Concern** (5jJk Q2): ``Why does Rank-2 accuracy fall below 50%?``

**Response Summary**:

Evaluation requires strict format matching (e.g., [[A>B>C]]). Malformed outputs → score = 0.
Random-choice accuracy assumes perfect formatting—an upper bound, not an empirical baseline.

**Revisions**:

- Clarified in Table 1 footnote (Lines 326–328) and added failure examples.

---

**(e) Suspicious 0% Accuracy Claim**

**Concern** (ahDM W3): ``Llama-3.3-70B showing 0% seems implausible.``

**Response Summary**:

Correction: This was a label swap in Fig. 2. The 0% belongs to Qwen3-8B, due to severe format collapse at 128K (repetitions, truncations, hallucinations).

**Revisions**:

- Fixed in revised Fig. 2 (Lines 108–119) and clarified in text.

---

> ### Author Response · Authors · 2025-11-29
> **Summary for the New Area Chair (Part II)**
>
> **(f) Motivation & Necessity of Long-Context Reward Models**
>
> **Concern** (7E75 W4): ``Whether improving reward models for long contexts is necessary, given that modern LLMs already support extended context windows``
>
> **Response Summary**:
>
> Long-context LLMs ≠ strong RMs (e.g., Llama-3.3-70B still fails as RM); cost/latency of using 70B+ models is prohibitive; process-aware evaluation (e.g., agent trajectories) demands specialized LongRMs.
>
> **Revisions**:
>
> - Preliminary Fig. 2 (Lines 108–119); added discussion in rebuttal citing [13–16].
>
> ---
>
> **(g) Generalization, Bias, and Task Coverage**
>
> **Concern** (5jJk W1, W2, Q1, Q4): ``Why does performance degrade on discriminative RMs (e.g., Math)? Does the method introduce length bias? Would trimming context before evaluation improve short-RM performance? Does response length (short-chosen-long-rejected vs. short-rejected-long-chosen) affect alignment?``
>
> **Response Summary**:
>
> Our approach targets generative reward models (GenRMs); the performance drop on discriminative RMs (DisRMs)—especially in Math—stems from (i) their high sensitivity to data scale/distribution and (ii) the lack of task-specific synthetic data. DisRM results are included only as a limited ablation. To mitigate length bias, we mix short-context data during training, though smaller models (8B) face a capacity trade-off between short- and long-context performance, while larger models (14B) balance both more effectively(Fig. 11a, Line 1134-1149). Controlled experiments confirm that response length (short-chosen-long-rejected vs. short-rejected-long-chosen) has a negligible impact on alignment quality (Response to Reviewer 5jJk (Part II)). Finally, trimming “redundant” context before using a short-context RM is impractical—it requires gold labels for redundancy detection, which are unavailable at inference, introducing train-test inconsistency and risking loss of critical information.
>
> **Revisions**:
>
> - DisRM degradation discussed in Table 7 section (Lines 1188–1196);
> - Length bias and capacity trade-offs analyzed in Appendix D.2 (Lines 1126–1133) and Fig. 11a (Lines 1134–1149);
> - Response-length robustness validated in Fig. 10 (Lines 1103–1124);
> - Context-trimming issue clarified in Section 4.2 (Lines 292–294) and Fig. 5.
>
> ---
>
> **(h) Artifact Release & Reproducibility**
>
> **Concern** (ahDM): ``No public URL yet.``
>
> **Response Summary**:
>
> Committed to full release post-acceptance (dataset, code, leaderboard, checkpoints) per ICLR double-blind policy.
> Example data and core scripts have been included in the supplementary material.
>
>
> ---
>
> ### **3. Summary of all manuscript revisions**
>
> - New experiments (Fig. 10–11, ablations);
> - Clarified methodology (Section 2.1-2.2, Section 4.1–4.3, Appendix A/D);
> - Corrected errors (Fig. 2);
>
> ---
>
> Thank you again for taking the time to review our manuscript and referring to our discussion with the reviewer!
>
> Best,
>
> Authors

---

### Meta-Review · Area_Chair_wnVS · 2026-01-10

**Summary:**

This paper introduces Long-RewardBench, a benchmark for evaluating reward models under long-context settings (up to 128K), and proposes a multi-stage pipeline to train LongRMs using cold-start SFT followed by preference optimization (DPO) with consistency/majority-voted synthetic supervision. Reviewers agree the problem is timely and the empirical results are generally strong for generative reward models, showing notable recovery of long-context preference accuracy. The rebuttal clarifies several points (metric mapping, why rank-2 can be below the theoretical random baseline due to strict formatting, and fixes a figure label swap that caused a confusing 0% result) and adds ablations on two-stage training and length-related effects.

Pros
1. Timely and practically relevant: long-context reward modeling and evaluation for agentic / trajectory-style settings is an important gap, and the benchmark direction is valuable.
2. Clear empirical signal for GenRMs: results suggest the proposed recipe can substantially improve long-context robustness, sometimes making smaller models competitive with much larger ones.
3. Rebuttal meaningfully improved clarity: added task-to-metric table, corrected an apparent error in a key figure, and provided targeted experiments (SFT token scaling, response-length controls, short/long data mixing) that address several reviewer questions.

Cons
1. Ground-truth validity remains a core risk: although the authors clarify that judgments come from task-specific automatic metrics (and LLMs generate explanations), the benchmark still inherits the limitations of those proxies (e.g., ROUGE/heuristic scores) for capturing preference in many long-form tasks, and this is central to the paper’s claims.
2. Contribution novelty is somewhat incremental: the training recipe (SFT then DPO with voting-based data synthesis) is plausible and well engineered, but remains largely procedural, with limited new conceptual framing beyond tailoring to observed failure patterns.
3. Generalization and trade-offs are not fully resolved: performance degradation on discriminative RMs (notably math) and residual length-bias/capacity trade-offs suggest the approach may not be uniformly reliable; the paper positions DisRM results as ablations, but they still weaken the breadth of the claimed impact.

The paper addresses an important problem and the rebuttal resolves multiple concrete issues, but the remaining uncertainty around benchmark “ground-truth” faithfulness to human preference, plus the reliance on a largely procedural recipe and unresolved generalization trade-offs (especially beyond GenRMs), makes the contribution feel slightly short of the bar this cycle. I would be open to acceptance if the final version strengthened the validity story (e.g., targeted human checks or stronger justification of metric adequacy per task), clarified remaining failure modes, and provided a concrete, verifiable artifact/release plan that reviewers can trust.

**Reviewer Concerns:**

1. Why SFT then RL (two-stage) instead of only SFT or direct RL
They added clearer motivation tied to format failures, plus ablations showing SFT saturation / degradation and why RL helps after format learning.

2. What automatic metrics are used / whether ROUGE/F1/etc. are defined clearly
They provided a full task-to-metric mapping table and clarified the evaluation pipeline wording (judgment from metrics, LLMs only for explanations).

3. Pairwise vs pointwise + why scalar scoring / majority voting helps
They explained the “divide-and-conquer” pointwise decomposition, added rubric figures, and argued majority voting reduces instability.

4. Rank-2 below random and format violations
They clarified random-choice is a theoretical upper bound assuming perfect formatting, and showed concrete malformed-output examples; also updated the table note.

5. Suspicious 0% result for Llama-3.3-70B
They acknowledged and fixed a figure label swap and provided examples showing the true 0% case is driven by format collapse in the smaller model.

6. Length bias / short-vs-long preference artifacts
They added controlled experiments (short-chosen vs long-chosen vs balanced) and data-mixing ablations showing the effect is small and capacity-dependent.

**Reviewer Scores:**

N/A

---

### Decision · Program_Chairs · 2026-01-26

Reject